# Development and evaluation of habitat suitability models for nesting white-headed woodpecker (*Dryobates albolarvatus*) in burned forest

**Quresh S. Latif**[1¤]*, **Victoria A. Saab**[1]*, **Jonathan G. Dudley**[2], **Amy Markus**[3], **Kim Mellen-McLean**[4]

1 Rocky Mountain Research Station, U. S. Forest Service, Bozeman, Montana, United States of America,
2 Rocky Mountain Research Station, U. S. Forest Service, Boise, Idaho, United States of America,
3 Fremont-Winema National Forest, U. S. Forest Service, Lakeview, Oregon, United States of America,
4 Pacific Northwest Region (Region 6), U. S. Forest Service, Oregon city, Oregon, United States of America

¤ Current address: Bird Conservancy of the Rockies, Fort Collins, Colorado, United States of America
* quresh.latif@birdconservancy.org (QSL); vsaab@usda.gov (VAS)

**Data Availability Statement:** All relevant data are within the manuscript and Supporting Information files.

## Abstract

Salvage logging in burned forests can negatively affect habitat for white-headed woodpeckers (*Dryobates albolarvatus*), a species of conservation concern, but also meets socioeconomic demands for timber and human safety. Habitat suitability index (HSI) models can inform forest management activities to help meet habitat conservation objectives. Informing post-fire forest management, however, involves model application at new locations as wildfires occur, requiring evaluation of predictive performance across locations. We developed HSI models for white-headed woodpeckers using nest sites from two burned-forest locations in Oregon, the Toolbox (2002) and Canyon Creek (2015) fires. We measured predictive performance by developing one model at each of the two locations and quantifying discrimination of nest from reference sites at two other wildfire locations where the model had not been developed (either Toolbox or Canyon Creek, and the Barry Point Fire [2011]). We developed and evaluated Maxent models based on remotely sensed environmental metrics to support habitat mapping, and weighted logistic regression (WLR) models that combined remotely sensed and field-collected metrics to inform management prescriptions. Both Maxent and WLR models developed either at Canyon Creek or Toolbox performed adequately to inform management when applied at the alternate Toolbox or Canyon Creek location, respectively (area under the receiver-operating-characteristic curve [AUC] range = 0.61–0.72) but poorly when applied at Barry Point (AUC = 0.53–0.57). The final HSI models fitted to Toolbox and Canyon Creek data quantified suitable nesting habitat as severely burned or open sites adjacent to lower severity and closed canopy sites, where foraging presumably occurs. We suggest these models are applicable at locations similar to development locations but not at locations resembling Barry Point, which were characterized by more (pre-fire) canopy openings, larger diameter trees, less ponderosa pine (*Pinus ponderosa*), and more juniper (*Juniperus occidentalis*). Considering our results, we recommend

**Funding:** Funding was provided primarily by the United States Forest Service: Fremont-Winema National Forest (https://www.fs.usda.gov/fremont-winema/), Malheur National Forest (https://www.fs.usda.gov/malheur/), Rocky Mountain Research Station's (https://www.fs.usda.gov/rmrs/) National Fire Plan (02.RMS.C2), and the Pacific Northwest Region (https://www.fs.usda.gov/r6). Co-authors include current or former employees of funding agencies who also contributed to sampling design at one or more locations. Bird Conservancy of the Rockies supported QSL during manuscript preparation, submission, and peer review. The funders had no role in study design, data collection and analysis, decision to publish, or preparation of the manuscript.

**Competing interests:** The authors have declared that no competing interests exist.

caution when applying HSI models developed at individual wildfire locations to inform post-fire management at new locations without first evaluating predictive performance.

## Introduction

Wildfire influences vegetation structure and composition in dry conifer forests of western North America along with associated biological communities. Many species colonize recently burned forests, where resources generated by wildfire allow populations to proliferate [1, 2]. In particular, woodpeckers and other cavity-nesting species benefit from trees that are killed, injured, or weakened by fire for nesting and foraging [3–5]. Anthropogenic land use and climate change strongly influence wildfire, fire-related ecological processes, and consequently habitat for fire-associated species [6, 7].

Salvage logging in particular negatively impacts fire-associated species by targeting a key resource upon which they depend: relatively large trees killed by fire [4, 5, 8, 9]. Despite broad evidence for primarily negative impacts on biodiversity, managers commonly apply salvage logging to recoup economic loss of timber resources and mitigate human safety hazards following wildfire [10, 11]. Increased size and severity of wildfire with warming temperatures [6, 7] may further increase opportunity and perceived socioeconomic need for salvage logging. Forest managers must therefore balance socio-economic demands with mandates requiring maintenance of post-fire habitat for wildlife.

Researchers use habitat suitability models (sometimes known as species distribution models) to identify suitable habitat and predict species distributions to inform land management decisions aimed at species conservation [12]. These models quantify environmental relationships with known species occurrences, and based on these, predict species distribution. Models often provide habitat suitability indices (HSIs; 0–1 range) that at minimum indicate relative likelihood of species occurrence (0 = least likely, 1 = most likely). Interpretation of HSIs and their value for ecological inference is the subject of ongoing debate and depends at least in part on modeling technique and data used for model development [13–17]. Nevertheless, to inform habitat conservation, models are ultimately expected to discriminate where species are most, versus least likely to occur within relevant areas.

How we develop and evaluate habitat models must reflect both species ecology and intended applications. Often predictive maps that provide continuous and broad coverage are needed to inform conservation and management planning. Models developed for these purposes typically employ environmental variables derived from remotely sensed data [18–20], which are typically coarse in resolution and information content [21]. Thus, restricting models to remotely sensed data can limit performance by limiting their ability to quantify key relationships governing species distributions at finer resolutions [3]. Consequently, including field-measured data can improve performance [3]. Incorporating field-collected data may preclude habitat mapping over broad spatial extents, but may provide finer resolution information useful for management prescriptions to maintain or improve habitat suitability.

Regardless of the particular application, models informing habitat management for fire-associated species must continually be applied to new locations as new wildfires occur [e.g., 22]. Many factors can cause geographic variability in model applicability, however, including biotic interactions, local adaptation, and behavioral rules governing habitat selection [23–26]. Because of funding limitations and the unpredictability of wildfire, models for disturbance-associated woodpeckers typically represent individual wildfire locations [3, 5, 27, except see

28], potentially limiting wider model applicability. Such concerns are common for predictive habitat models [23, 24, 29], raising the need to evaluate applicability across locations to test transferability [ability of models to provide useful predictions when applied at new locations beyond where they were originally developed; [30–32]. Models that consistently describe occurrence patterns throughout a species range can be generally applied to informing management. Conversely, given spatial variability in environmental relationships, comparing predictions across locations can reveal limitations to model applicability [32].

The white-headed woodpecker (*Dryobates albolarvatus*) is a fire-associated species of conservation concern. The species is endemic to dry, conifer forests of western North America, where habitat loss and degradation due to human activities and consequent alteration of fire regimes has raised conservation concerns [33]. White-headed woodpeckers nest in both recently burned and unburned forests typically within landscapes characterized by mosaics of open- and closed-canopy forests often generated and maintained by mixed-severity fire [27, 34, 35]. In burned forests, nests are typically placed in moderate-to-severely burned or open-canopied sites adjacent to less-burned or closed-canopy areas, which contain greater densities of live trees thought to provide food resources [27]. Additionally, nest cavities are typically excavated in decayed snags within forest stands dominated by ponderosa pine (*Pinus ponderosa*) in the northern portion of their range. Nest survival can be substantially higher in burned compared to unburned forests, suggesting burned forests could represent source habitat for maintaining populations [27, 34]. Therefore, conservation of burned forest habitats may be particularly important for population persistence, but complex environmental relationships make identifying suitable nesting habitat challenging.

Here, we developed and evaluated habitat suitability models for nesting white-headed woodpeckers in burned forests. Our objectives were 1) to develop models capable of supporting both coarse-resolution habitat mapping and management prescriptions, 2) to evaluate models by testing their transferability across wildfire locations, thereby establishing their broad applicability to inform post-fire forest management.

## Materials and methods

### Ethics statement

All fieldwork for this study was conducted on public lands. None of our study species were listed as threatened or endangered under the U.S. Endangered Species Act. Data collection was purely observational and did not involve any physical contact with study organisms. All work was conducted following best practices for minimizing observer impacts on nesting birds [36].

### Study locations and management

We modeled habitat relationships for nesting white-headed woodpeckers at the Toolbox and Silver fire complex in central Oregon (hereafter the Toolbox Fire; 42˚57´N, 120˚59´W) and at the Canyon Creek Complex in eastern Oregon (hereafter Canyon Creek Fire; 44˚17´N, 118˚51´W). We evaluated model predictions by applying models between these and a third location, the Barry Point Fire in southern Oregon (42˚04´N, 120˚39´W; Table 1, Fig 1). Preliminary analysis revealed different relationships at Barry Point compared to the other two locations, resulting in poor predictive performance when applying Barry Point models at Toolbox and Canyon Creek (V. Saab unpublished data). Considering the limited sample size at Barry Point (*n* = 19 nest sites), we lacked confidence in relationships observed at Barry Point for contributing meaningfully to general knowledge of white-headed woodpecker nesting habitat relationships. We therefore abandoned model development at Barry Point for this study and used Barry Point data exclusively to evaluate models developed at the other two locations. Before

**Table 1. Timing, size, and sampling of three Oregon wildfires where white-headed woodpecker nests were located to develop and evaluate habitat suitability models.**

| National Forest | Fire Name | Ignition Year | Years surveyed | Full extent (ha) | Surveyed extent (ha) | No. pixels with nests[a] |
|---|---|---|---|---|---|---|
| Fremont-Winema | Toolbox | 2002 | 2003–2007 | 33,427 | 856[b] | 46[b] |
| Fremont-Winema | Barry Point | 2011 | 2012 | 12,352[c] | 1,603 | 19[d] |
| Malheur | Canyon Creek | 2015 | 2016–2017 | 44,672 | 4,347, 4,727[e] | 47 |

[a]We treated each pixel containing ≥ 1 nest as one observation for habitat models. We located 47 nests at Toolbox, but two were located in the same pixel.

[b]Non-nest sites were only measured in the 13 larger of 22 survey units. In these 13 units, area surveyed = 798 ha and 33 nests were located.

[c]The Barry Point Fire extended into California, but only the Oregon extent is represented here.

[d]Barry Point data were used for model evaluation but not development because relationships differed from other study locations, and we questioned the generality of Barry Point relationships considering the limited sample size.

[e]One survey unit at Canyon Creek was replaced between years; area surveyed was 4,347 ha in 2016 and 4,727 ha in 2017.

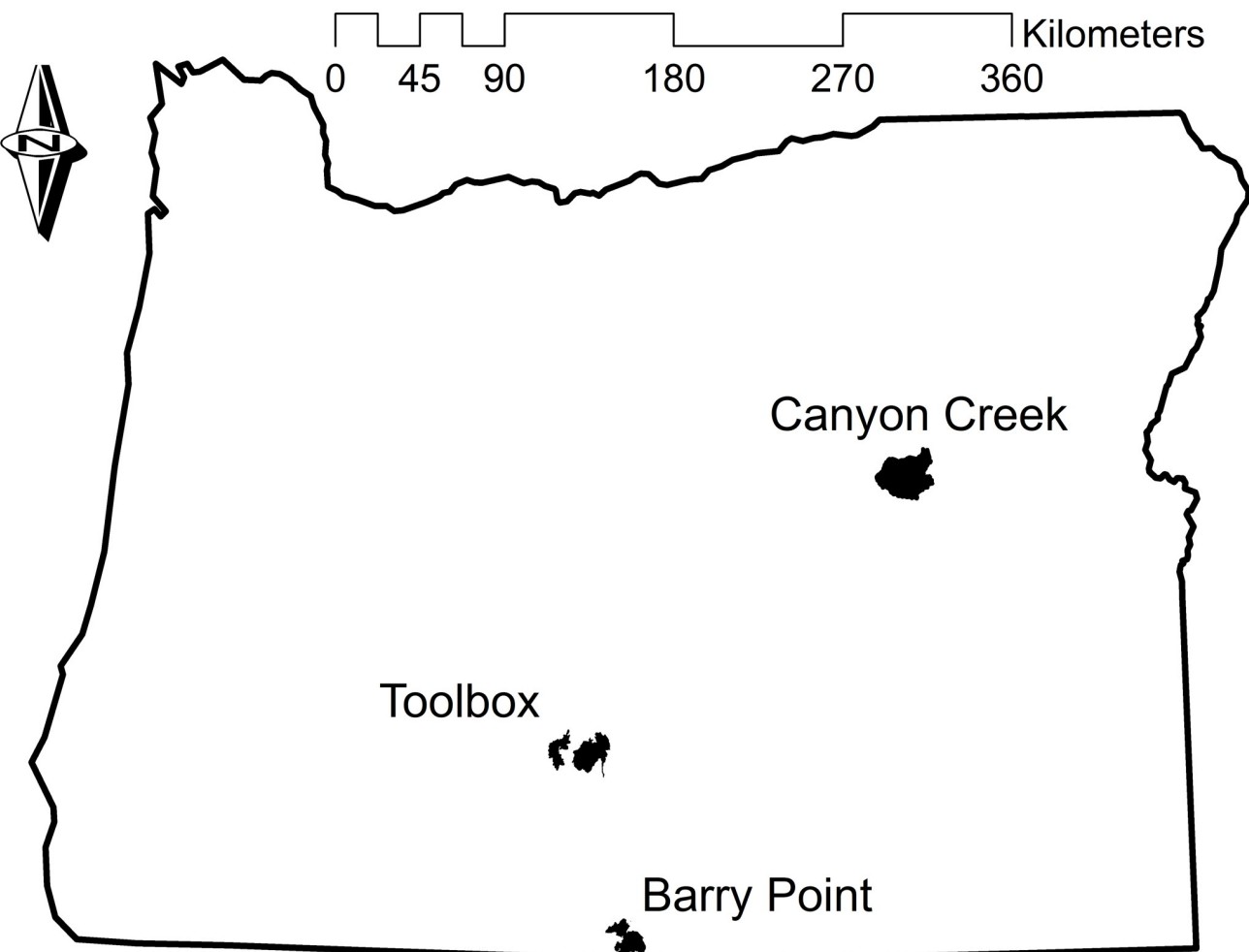

**Fig 1. Wildfire locations.** Maps showing the three study locations where habitat suitability models were developed and evaluated for white-headed woodpeckers in burned forests, Oregon, U.S.A.

wildfire, all three study locations were characterized as dry mixed-conifer forest, much of which was dominated or co-dominated by ponderosa pine and managed for multiple uses (e.g., wildlife habitat, timber harvest, and grazing).

Salvage logging was implemented in portions of the Toolbox and Canyon Creek Fires. Lands within both fire perimeters were owned by a mix of public (U.S. Forest Service [USFS], Bureau of Land Management, and the state of Oregon) and private entities. Most private property containing merchantable timber was logged immediately following wildfire. Logging activities on USFS lands focused on *a priori* identified sale units located within a subset of our survey units (described below). Logging prescriptions implemented at the Toolbox Fire retained ≥ 25 snags per hectare of diameters representing the range of pre-treatment tree and snag diameters, and retained snags were distributed in clumps of ≥ 100 snags per 4 hectares [37]. Except for immediately logged private lands, salvage logging was primarily implemented in autumn of 2004 at Toolbox, resulting in increased total logging extent from 2004 to 2005 breeding seasons (375 to 5,946 ha) and a smaller increase from 2005 to 2007 (6,249 ha) within 1 km of study units. At the Canyon Creek Fire, roadside salvage logging was implemented in 2016 before, during, and after nest surveys, wherein 799 ha (283 ha within our survey units) along roads were cleared of standing dead trees identified as hazardous to the public [38]. Between the 2016 and 2017 breeding seasons, additional selective harvest was implemented across 490 ha of surveyed areas, wherein prescribed retention was 84–126 snags of diameter at breast height (DBH) > 23 cm per hectare. Salvage logging was not implemented at the Barry Point Fire. Although wildfire locations included some private land, we restricted surveys to public land.

### Nest surveys and reference sites

We surveyed rectangular belt transects spanning *a priori* established survey units (Table 1, Fig 1) to locate occupied nest cavities [36] during early May until mid-July 1–5 years following wildfire (Table 1). We placed the center of belt transects 200 m apart and surveyed 100 m on either side of the center line. Transects began and ended at opposing unit boundaries, so surveys covered each unit. We surveyed 22 units at Toolbox (2–116 ha), 9 units at Canyon Creek (161–403 ha), and 5 units at Barry Point (164–439 ha). At Barry Point and Canyon Creek, survey units included suitable and unsuitable habitat identified by a preliminary model developed with white-headed woodpecker nest site data from Toolbox (V. Saab unpublished data). We incorporated call broadcasts into our surveys to elicit responses by territorial woodpeckers and thereby improve detection. We used GPS units (Garmin Etrex, Garmin International, Inc, Olathe, KS 66062; Trimble GeoExplorer3, Trimble Navigation Limited 1999–2001, Sunnyvale, CA 94085) to determine the geographic coordinates of each nest cavity. Surveyors typically remained within survey unit boundaries but occasionally strayed up to 250 m outside unit boundaries when following specific individuals exhibiting signs of breeding behavior to locate a nest cavity. Thus, nest sites were occasionally located just outside unit boundaries.

Habitat suitability models compared environmental conditions at nest to reference sites. For models restricted to remotely sensed data, reference sites were 10,000 30-m pixels drawn randomly from within the area surveyed at each wildfire location (hereafter available sites). For models developed with remotely sensed and field-collected data, reference sites were 134, 176, and 21 non-nest sites randomly located within survey unit boundaries at Toolbox, Canyon Creek, and Barry Point locations, respectively. We centered non-nest measurements on the tree nearest to each randomly generated coordinate and then re-measured non-nest coordinates in the field with GPS units at the tree where measurements were centered. All non-nest sites were ≥ 35 m away from the nearest nest site located during the study period. Given the

high detectability of cavity nests especially during the nestling period [39] and the high survival rate of white-headed woodpecker nests following wildfire [27], we assumed low likelihood of undetected active nests at non-nest sites.

## Environmental data at nest and reference sites

We compiled five remotely sensed and five field-collected environmental variables for use as modeling covariates, along with additional salvage logging metrics compiled solely to inform discussion (Table 2). Remotely sensed data, compiled at a 30-m-pixel resolution, described topography, pre-fire canopy cover, burn severity, and extent of ponderosa pine-dominated forest at all wildfire locations. We also compiled salvage logging extent at Toolbox and Canyon Creek locations. Biological relevance of these variables is described by previous authors [5, 27, 34, 35]. Remotely sensed variables described either a local (single pixel or a 9-pixel [0.81-ha] neighborhood) or a landscape scale [3,409-pixel [1-km radius; 314-ha] neighborhood; approximate area likely containing a home range; 33]. We derived topographic variables from LAND-FIRE [40]. Previous studies described associations with topographic slope and aspect, which can influence microclimate and consequent vegetation structure [27, 34, 35]. We quantified burn severity using data from Monitoring Trends in Burn Severity [41] and canopy cover using Gradient Nearest Neighbor (GNN) data [imagery year 2002; 42]. We used the delta normalized burn ratio index [$\Delta$NBR, 43] to identify moderate-to-severely burned pixels [$\Delta$NBR > 270; following 27], and we assumed pixels classified as "non-forest" to have zero pre-fire canopy cover. White-headed woodpeckers nest in canopy openings either generated by recent wildfire or present before fire [27, 34, and 35]. We expected white-headed woodpeckers to include some unburned or low-severity burned forest with closed canopies in their home ranges for foraging. We therefore modeled habitat suitability as a function of the proportion of area burned ($\Delta$NBR > 270) or open (canopy cover < 10%) at the nest site (0.81 ha) and home range (314 ha) scales (inter-scale correlations were $r$ = 0.46, -0.05, and 0.48 at Toolbox, Barry Point, and Canyon Creek, respectively [$n$ = 10,000, 10,000, and 20,000 pixels, respectively]). We expected habitat suitability to relate positively with burned or open canopies at the nest-site scale while also relating negatively with burned or open canopies at the home-range scale when accounting for relationships at both scales in the same model. We assigned values to nests reflecting the year in which they were found and we averaged values across years for each non-nest point such that non-nest data reflected the average conditions available over spatial and temporal extents surveyed.

Field-collected variables described either characteristics of 50-m radius patches or individual nest and non-nest snags (i.e., local scale descriptors of the nest site; Table 2). The specific dimensions of sampled plots used to measure tree and snag densities varied somewhat among study locations (S1 Appendix) and were therefore rescaled to represent per hectare counts for analysis. We also recorded the size, species, and status (live versus dead) of nest and non-nest trees. Both previous research [27, 34] and data collected here indicate white-headed woodpeckers nest almost exclusively in snags with DBH $\geq$ 25 cm. White-headed woodpeckers can nest in dead portions of live trees, but we rarely observed this behavior at our study locations (1 nest at Barry Point). Therefore, we only considered snags with DBH $\geq$ 25 cm as non-nest trees (i.e., available but unused). If the center tree for a non-nest site was alive or too small (DBH < 25 cm), we randomly selected a snag $\geq$ 25 cm DBH located within the 50-m patch to represent the non-nest tree for that site. We excluded from analysis those non-nest points that lacked any snags $\geq$ 25 cm DBH within 50 m (3, 3, and 2 non-nest sites excluded at Toolbox, Canyon Creek, and Barry Point, respectively). Given these restrictions, we did not consider tree size or live/dead status as modeling covariates, although we do report DBH descriptive

**Table 2. Remotely sensed (remote) and field-collected (field) environmental variables measured at burned forest locations where habitat models were developed for nesting white-headed woodpeckers.**

| Variables (abbrev) | Type | Description | Modeling covariate? |
|---|---|---|---|
| Slope | remote | pixel topographic slope as % rise over run | yes |
| Cosine aspect (Casp)[a] | remote | pixel cosine-transformed (north-south) orientation of topographic slope | yes |
| Local-scale percent area burned or open (LocBrnOpn) | remote | Percentage of 3×3 cell (0.81 ha) neighborhood moderately to severely burned ($\Delta$NBR > 270) or <10% pre-fire canopy cover | yes |
| Landscape-scale percent area burned or open (LandBrnOpn) | remote | Percentage of 1-km radius (314 ha) neighborhood moderately to severely burned ($\Delta$NBR > 270) or <10% pre-fire canopy cover | yes |
| Landscape-scale percent area ponderosa pine forest (LandPIPO) | remote | Percentage of 1-km radius (314 ha) ponderosa pine forest[c] | yes |
| Local-scale extent of logging (LocLog)[b] | remote | Percentage of 3×3 cell (0.81 ha) neighborhood intersecting sale units for salvage logging. | no |
| Landscape-scale extent of logging (LandLog)[b] | remote | Percentage of 1-km radius (314 ha) neighborhood intersecting sale units for salvage logging. | no |
| Medium snag density (SngMidDens) | field | Number of medium snags (25–50 cm DBH) per ha within 50 m | yes |
| Large snag density (SngLrgDens) | field | Number of large snags (>50 cm DBH) per ha within 50 m | yes |
| Medium-to-large live tree density (TreeDens) | field | Number of medium-to-large trees (>25 cm DBH) per ha within 50 m | yes |
| Percent ponderosa pine (PIPO%) | field | Percentage of medium-to-large snags and trees (>25 cm DBH) that are ponderosa pine | yes |
| Ponderosa pine (PIPO) | field | Whether or not nest or center tree was ponderosa pine (categorical; 0 = no, 1 = yes) | yes |
| Logging intensity (LogIntensity)[b] | field | Ratio of cut stump density to density of all stumps, snags, and trees | no |

[a]Casp = 0 wherever Slope $\leq$ 2%.

[b]Only assessed at Toolbox and Canyon Creek locations to measure extent (LocLog, LandLog) and intensity (LogIntensity) of post-fire salvage logging. Logging variables were compiled for reference when interpreting modeling results but not used as modeling covariates. Additionally, the size distribution for cut stumps (< 1.4 m high, $\geq$ 25 cm top diameter), snags (dead, $\geq$ 1.4 m high, DBH $\geq$ 25 cm), and trees (alive, $\geq$ 1.4 m high, DBH $\geq$ 25 cm) were not equivalent because diameter was measured at different heights (< 1.4 m for stumps, at 1.4 m for trees and snags), so LogIntensity represents a relative index rather than an absolute measure of logging intensity.
[c]Ponderosa pine forest was defined based on forest type classifications provided with gradient nearest-neighbor data as all pixels listed as dominated or co-dominated by ponderosa pine [42].

statistics to inform discussion. Many woodpecker species favor decayed snags for nest cavity excavation [27, 44], but we did not record decay at Canyon Creek so we did not model relationships with decay. We measured non-nest points in the field concurrently with nest site measurements. Some non-nest points were measured repeatedly in multiple years ($n$ = 86 at Toolbox), in which case we used the mean of replicate measurements for model development.

In addition to environmental variables, we compiled metrics describing the extent and intensity of salvage logging at Toolbox and Canyon Creek at relevant spatial scales remotely and in the field (Table 2). We initially included logging covariates in models, but doing so reduced predictive performance (i.e., models discriminated nest from non-nest sites no better than random with logging covariates–area under the receiver-operating-characteristic curve [AUC] $\approx$ 0.5), perhaps reflecting variation in prescriptions and consequent implications of salvage logging among study locations. Regardless, we were mainly interested in informing management decisions prior to logging (e.g., designation of habitat reserves). We therefore only report descriptive statistics for logging metrics to inform inference from modeling results.

## Habitat suitability models

**Maxent with remotely sensed data.** To support habitat mapping, we developed Maxent models with remotely sensed data to differentiate environmental conditions at used (nest) versus available sites. Maxent is informed by use-availability data [a.k.a. presence-background; 45–47] and was found effective for quantifying habitat in unburned forest [35]. We used the

logistic Maxent output (0–1 range) as HSIs [see 35, 46]. Available sites informing Maxent models were 10,000 pixels for each wildfire location drawn randomly from within survey units and up to 250 m outside unit boundaries. After verifying comparable performance with more complex models, we favored relatively simple models to facilitate interpretation of habitat relationships (S2 Appendix). Accordingly, reported models only included variables with contributions of $\geq$ 5% gain in initial model runs [variable contributions described by 46]. Additionally, we only considered linear, quadratic, and interactive covariate effects [48].

**Weighted logistic regression with remotely sensed and field-collected data.** To inform management prescriptions, we developed weighted logistic regression (WLR) models informed by remotely sensed and field-collected data. We weighted non-nest sites ($y = 0$) and nest sites ($y = 1$) to negate the influence of their respective sample size on the estimated response [$w_1 = 1$; $w_0 = n_1/n_0$; 3, 32]. This scheme correctly treats the overall ratio of nest-to-non-nest sites as an artifact of sampling. The estimated response is thereby interpretable as a relative index of habitat suitability [HSI; 3]. To maximally inform discrimination of suitable from unsuitable habitat, zeros should represent unused sites uncontaminated with misclassified nest sites [49]. Our field methods resulted in a thorough search of study units, so we are reasonably confident that nests were never located within 30 m of non-nest sites (resolution of remotely sensed data) during the study period. We fitted weighted logistic regression models using the glm function in R [v. 3; 50]. Considering our sampling methods, we suspect WLR HSIs primarily quantified suitability for nest site selection. Because some nests were found after initiation, however, our data could additionally represent nest predation and competition, which also potentially shape white-headed woodpecker nesting distributions.

We constructed and compared candidate models with alternative covariate combinations using an information theoretic framework [51]. We constructed candidate models describing all combinations of relevant covariates limited to a maximum of 1 covariate per 10 nests rounded up (i.e., 5 covariates for $n = 46$ and 47 nests at each of Toolbox and Canyon Creek locations, respectively) to avoid overfitting. We only considered first-order linear covariate relationships and did not consider quadratic, interactive, or higher order effects. We compared candidate models using small-sample corrected Akaike's Information Criterion (AIC$_c$) and AIC$_c$ model weights. We first fitted models to individual study locations with sufficient nest site data to support model development (Toolbox, Canyon Creek) and retained top models (lowest AIC$_c$) for evaluating predictive performance. Variance inflation factors [see 52] for all covariates were $\leq$ 2.57 (i.e., $R^2 \leq 0.61$ when regressing a given covariate against all other covariates), so multicollinearity was not a concern. At the Toolbox location, we only measured field-collected variables at non-nest sites within the 13 largest survey units ($\geq$ 23.3 ha), so WLR models were fitted to data from these units (33 nest and 134 non-nest sites).

## Model evaluation

We assessed transferability of models developed at individual wildfire locations (Toolbox and Canyon Creek) by evaluating predictions applied at alternate locations. We measured predictive performance using AUC [53] to measure discrimination accuracy of nest from non-nest sites. An AUC = 0.5 indicates discrimination no better than random, whereas AUC = 1 indicates perfect discrimination [53]. We considered model predictions useful for discriminating nest from reference sites when the lower limit for the 95% confidence interval (CI) for AUC exceeded 0.5. We used the pROC package in R to calculate bootstrapped AUC CIs [54].

We considered transferability indicative of consistency in environmental relationships across wildfire locations, so we pooled data across locations where models were transferable to develop a final model to inform management (hereafter *pooled models*). We re-ran model

selection and fitting procedures (see above) using covariates included in the final (Maxent) or top-ranked (WLR; within 2 AIC units) models at individual locations as candidate covariates when developing pooled models. So that pooled models would be informed equally by each individual wildfire location, we adjusted the analyzed data as follows. For the pooled Maxent model, the proportion of available (background) sites from each location was set to match the proportion of nest sites from each location. For the pooled WLR model, we down-weighted data from the location with a larger sample of nests (Canyon Creek) so that the sum of the weights for observations from each location equaled the sum of observation weights for the other location in the pooled dataset.

## HSI relationships with hatched-nest densities

We related HSIs with observed densities of hatched nests (i.e., nests with at least 1 nestling), reflecting both nest site selection and a component of fitness, nest survival to hatching. HSI relationships with hatched-nest densities can inform interpretation and application of HSIs in terms relevant to forest management and population targets. Additionally, hatched nests for wildfire-associated woodpeckers are highly detectable [39], reducing the need to account for detection probability when estimating densities. We verified hatching status by monitoring nests and checking their status regularly [36]. We plotted the density of hatched nests for equal-area moving window bins [described by 29] to visualize density changes with increasing HSI. Additionally, we used HSI relationships with hatched-nest densities to identify natural breaks useful for classifying suitability classes often desired for management planning [55]. We selected two thresholds to distinguish three potential suitability classes (low, moderate, and high suitability) that clearly differed in hatched-nest densities. Two nests at each of Toolbox and Canyon Creek locations did not hatch and were therefore excluded from samples used to relate HSIs with hatched-nest densities.

We calculated 95% CIs for hatched-nest densities within suitability classes defined by HSI thresholds using bootstrapping [56]. We used 600-m resolution cells forming a grid that extended across study units as sampling units for bootstrapping. We assigned nest, non-nest, and available sites the IDs of cells containing them, and we resampled the data by cell ID with replacement to generate 5,000 bootstrapped samples ($n = 57$ and 169 cells for Toolbox nest–non-nest and use-availability data, respectively; $n = 104$ and 212 cells for Canyon Creek nest–non-nest and use-availability data, respectively). We assumed non-nest (for WLR) and available (for Maxent) sites accurately represented the proportion of area surveyed in each HSI class for estimating class-specific densities. We report as confidence limits the 2.5% and 97.5% median-unbiased quantiles for bootstrapped samples calculated with the *quantile* function in R (type = 8).

We provide R scripts and an R workspace with data needed to replicate all analyses and plots in this manuscript (S1 Data).

## Results

Conditions at nest sites differed notably from non-nest sites at all wildfire locations (Table 3). Live tree densities (TreeDens) were consistently lower at nest compared to non-nest sites. Other notable patterns were not consistent across locations. For example, Toolbox and Canyon Creek nest sites were more severely burned or open (LocBrnOpn) at a local scale but less severely burned and less open at a landscape scale (LandBrnOpn) than non-nest sites. We did not observe this apparent scale-dependent tradeoff at Barry Point. At Barry Point, the extent of ponderosa pine-dominated forest (LandPIPO) at nest sites deviated more positively from non-nest sites than was apparent at Toolbox or Canyon Creek locations. Overall conditions

**Table 3. Mean (SD) values for remotely sensed and field-collected variables for nest and non-nest sites at three wildfire study locations.** Complete variable names and descriptions are in Table 2. n = 33 and 134 for Toolbox nests and non-nests, n = 47 and 176 for Canyon Creek nests and non-nests, and n = 19 and 19 for Barry Point nests and non-nests, respectively. Units are % for Slope, number per ha for tree and snag densities (SngMidDens, SngLrgDens, TreeDens), and cm for DBH.

| Variables | Toolbox | | Canyon Creek | | Barry Point | |
|---|---|---|---|---|---|---|
| | nest | non-nest | nest | non-nest | nest | non-nest |
| Slope[a] | 7.3(5.6) | 7.8(6.6) | 21.3(12.9) | 23.5(11.4) | 9.2(6.2) | 9.1(6.6) |
| Casp[a] | 0.19(0.66) | 0.27(0.57) | -0.16(0.7) | -0.18(0.69) | -0.34(0.54) | -0.09(0.68) |
| LocBrnOpn[a] | 95.3(13) | 81.6(32.4) | 82(26.8) | 80.4(28.9) | 77.2(29.5) | 73.1(33.6) |
| LandBrnOpn[a] | 61.1(19.7) | 65.7(21.5) | 60.7(14) | 68.1(13.7) | 69.8(7.4) | 69.3(10) |
| LandPIPO[a] | 74.9(7.9) | 72.3(10.6) | 59.8(10.1) | 59.2(10.5) | 51.7(5.7) | 31.5(28.2) |
| LocLog[a,e] | 24.9(41.9) | 19.2(36.9) | 13(30.1) | 18.4(33.9) | 0(0) | 0(0) |
| LandLog[a,e] | 29.6(21.2) | 21.6(22.8) | 13.2(11.8) | 13.2(13.5) | 0(0) | 0(0) |
| SngMidDens[b] | 65.1(39.2) | 57.6(42.3) | 93.6(54.2) | 83.9(51.7) | 63(41.8) | 74.3(41.4) |
| SngLrgDens[b] | 10.3(12.4) | 7.9(11.8) | 13.8(10.6) | 13.7(12.1) | 16.2(12.3) | 13.6(10.4) |
| TreeDens[b] | 5.3(13.5) | 26.8(35.6) | 27.8(57.2) | 63.4(115.3) | 14.2(21.6) | 23.7(28.5) |
| PIPO%[b] | 39(27.7) | 34.3(28.2) | 62.5(33.6) | 57(31) | 36.3(19.2) | 36.3(28) |
| PIPO[b,c] | 0.36 | 0.38 | 0.55 | 0.55 | 0.47 | 0.47 |
| DBH[b,d,e] | 35.7(16.9) | 37.7(16.7) | 47.3(19.8) | 41.5(14.6) | 59.2(20.4) | 47.8(25.3) |
| LogIntensity[b,e,f] | 0.21(0.26) | 0.09(0.2) | 0.09(0.17) | 0.11(0.2) | 0(0) | 0(0) |

[a]remotely sensed.

[b]field-collected.

[c]Categorical variables–reported values are proportion ponderosa pine.

[d]DBH = diameter breast height of nest or center snag.

[e]DBH and logging variables are described for reference but were not used for modeling.

[f]Logging conditions varied through time, and values represent conditions averaged across sites and years.

available for nesting also varied among locations (see non-nest sites, Table 3). Toolbox and Canyon Creek locations were characterized by greater coverage of ponderosa-dominated forest (LandPIPO), smaller trees (DBH), and less severely burned or less open canopies at a landscape scale (LandBrnOpn) than at Barry Point. Logging at Toolbox was more intense (LogIntensity) and more extensive (LocLog, LandLog) at nest compared to non-nest sites, whereas this pattern did not hold at Canyon Creek (Table 3).

Maxent models consistently retained local- and landscape-scale percent area burned or open (LocBrnOpn and LandBrnOpn) variables as primary contributors (Table 4). Maxent HSIs described positive and negative nest habitat relationships with LocBrnOpn and LandBrnOpn, respectively (Fig 2). LandPIPO also contributed to the Toolbox model and Slope to the Canyon Creek model (Table 4). At Toolbox, Maxent HSIs related positively with ponderosa-pine dominated forest, and at Canyon Creek, HSIs related negatively with topographic slope (Fig 2). These covariates represented relatively minor contributions, however, and were not retained in the pooled model (Table 4).

**Table 4. Variable contributions (% gain) for Maxent models developed with remotely sensed data measured at white-headed woodpecker nest and available sites in burned forest (Oregon, USA).** Models at individual locations (Toolbox, Canyon Creek) were evaluated for transferability before pooling (Toolbox & Canyon Creek).

| Variable | Toolbox | Canyon Creek | Toolbox & Canyon Creek |
|---|---|---|---|
| LocBrnOpn | 55.8 | 38.5 | 53.3 |
| LandBrnOpn | 38.6 | 47.8 | 46.7 |
| LandPIPO | 5.6 | -- | -- |
| Slope | -- | 13.7 | -- |

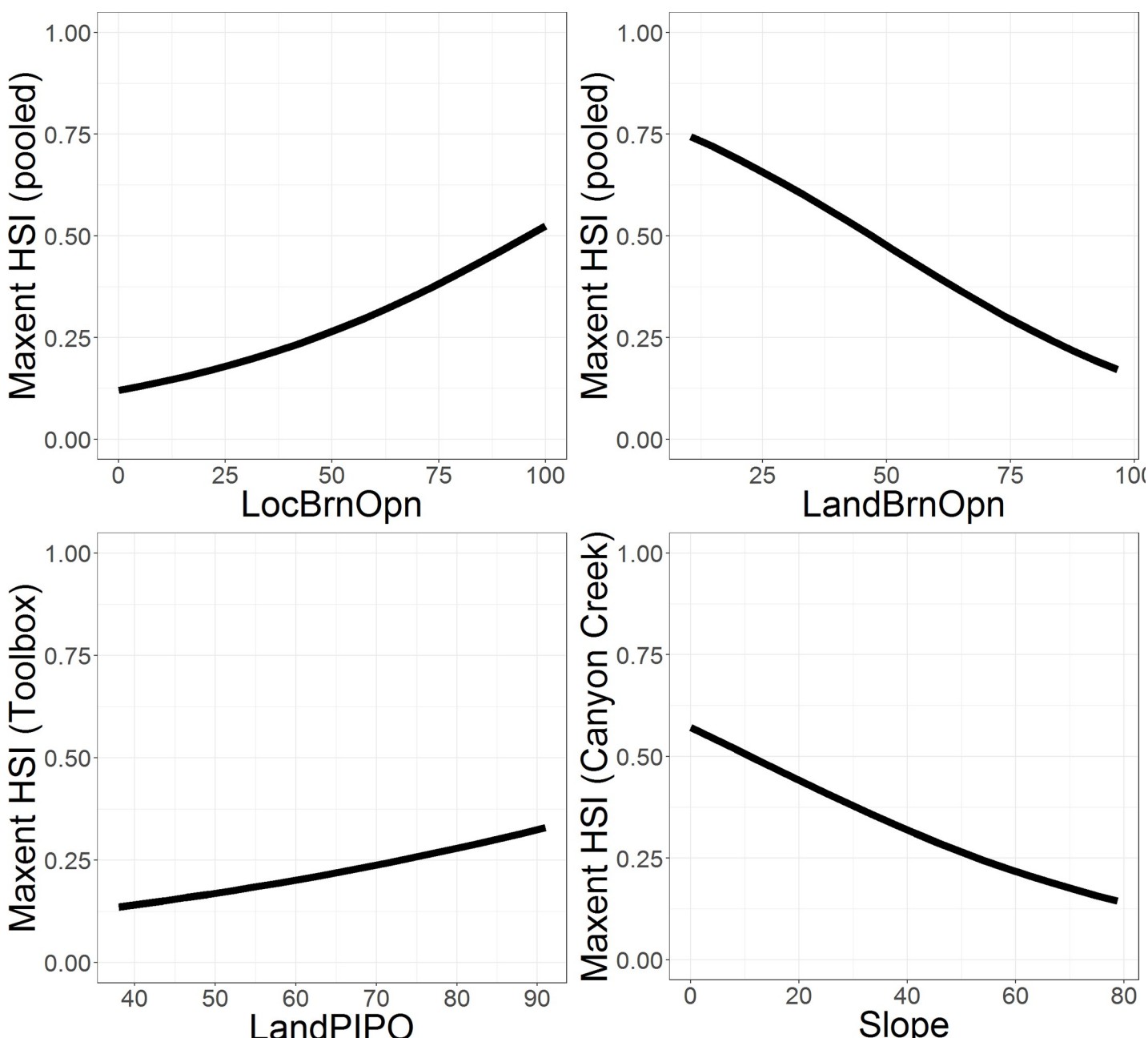

**Fig 2. Maxent HSI relationships with underlying covariates.** Covariates are local- and landscape-scale percent area burned or open (LocBrnOpn, LandBrnOpn), landscape-scale percent ponderosa pine-dominated forest (LandPIPO), and percent topographic slope (Slope). Complete descriptions are in Table 2. LandPIPO and Slope relationships (bottom panels) are from models developed at individual wildfire locations (Toolbox, Canyon Creek) and were not included in the final model intended to inform management (pooled model) but are reported to inform discussion.

Top-ranked WLR models described nest habitat relationships with burn severity or canopy openness (LocBrnOpn, LandBrnOpn), live tree density (TreeDens), and ponderosa pine (PIPO% or LandPIPO; Table 5). As with Maxent models, WLR HSIs related positively with LocBrnOpn and negatively with LandBrnOpn, again supporting a scale-specific tradeoff with burn severity and canopy openness (Fig 3, Table 6). WLR HSIs also related negatively with TreeDens (live tree density), additionally indicating selection for burned nest sites, and

**Table 5. Model selection results for weighted logistic regression models describing nest site selection by white-headed woodpeckers in burned forest.** Models within 2 AIC$_c$ units from the top-ranked (lowest AIC$_c$) model and the intercept-only model are presented. The total number of candidate models considered were 386, 638, and 64 for Toolbox, Canyon Creek, and pooled datasets, respectively. Complete lists of candidate models for each dataset are included in S1 Data. Complete covariate names and descriptions are in Table 2.

| Developed at | Covariates | K | ΔAIC$_c$ |
|---|---|---|---|
| Toolbox | LocBrnOpn + LandBrnOpn + TreeDens + PIPO% | 5 | 0.0 |
| | LocBrnOpn + LandBrnOpn + LandPIPO + TreeDens | 5 | 0.5 |
| | LocBrnOpn + LandBrnOpn + TreeDens | 4 | 0.8 |
| | LandBrnOpn + TreeDens + PIPO% | 4 | 1.4 |
| | Intercept-only | 1 | 14.0 |
| Canyon Creek | LandBrnOpn + TreeDens | 3 | 0.0 |
| | LandBrnOpn + TreeDens + PIPO% | 4 | 1.1 |
| | LocBrnOpn + LandBrnOpn + TreeDens | 4 | 1.3 |
| | LandBrnOpn + TreeDens + PIPO | 4 | 2.0 |
| | Intercept-only | 1 | 9.3 |
| Toolbox & Canyon Creek | LocBrnOpn + LandBrnOpn + TreeDens + PIPO% | 5 | 0.0 |
| | LandBrnOpn + TreeDens + PIPO% | 4 | 0.3 |
| | LandBrnOpn + TreeDens | 3 | 1.2 |
| | LandBrnOpn + LandBrnOpn + TreeDens | 4 | 1.8 |
| | LocBrnOpn + LandBrnOpn + LandPIPO + TreeDens + PIPO% | 6 | 1.9 |
| | Intercept-only | 1 | 10.6 |

[a]lowest AIC$_c$ = 80.4, 123.1, and 175.0 for Toolbox, Canyon Creek, and pooled models, respectively.

positively with PIPO% (i.e., dominance of nest sites by ponderosa pine). The top-ranked model at Canyon Creek excluded LocBrnOpn and PIPO%, indicating weaker nest habitat relationships with these variables at that location (Table 6). The negative estimated relationship with live tree density at Canyon Creek, however, suggests an affinity for locally burned or open nest sites consistent with patterns at Toolbox. The absence of PIPO% from the Canyon Creek WLR model mirrored the absence of LandPIPO from the Canyon Creek Maxent model, in contrast with Toolbox models, which consistently described positive nest habitat relationships with ponderosa pine variables.

Models developed at each of the Toolbox and Canyon Creek locations exhibited transferability between those two locations, but not to Barry Point (Table 7). Compared to development locations, AUC scores tended to be lower at alternate locations (except Canyon Creek Maxent model applied at Toolbox). AUC CIs consistently exceeded 0.5 at Canyon Creek and Toolbox locations, suggesting models remained informative there, whereas AUCs were consistently lower with 95% CIs that overlapped 0.5 at Barry Point. This pattern was consistent regardless of model type and data used for model development (i.e., Maxent informed by remotely sensed data versus WLR informed by remotely sensed and field-collected data). We therefore excluded Barry Point data from final pooled models (described in Tables 4, 5, 6, 7 and 8, and Figs 2 and 3).

Hatched-nest densities differed along HSI gradients and among suitability classes for pooled models at Canyon Creek and Toolbox locations (Table 8, Fig 4). Based on densities for moving-window bins, we identified Maxent HSI thresholds of 0.34 and 0.6 and WLR HSI thresholds of 0.3 and 0.53 for classifying suitability. Resulting categories of low, moderate, and high suitability habitat contained distinct hatched-nest densities, although for WLR HSI densities primarily differed in high suitability habitat compared to low and moderate (Fig 4). At

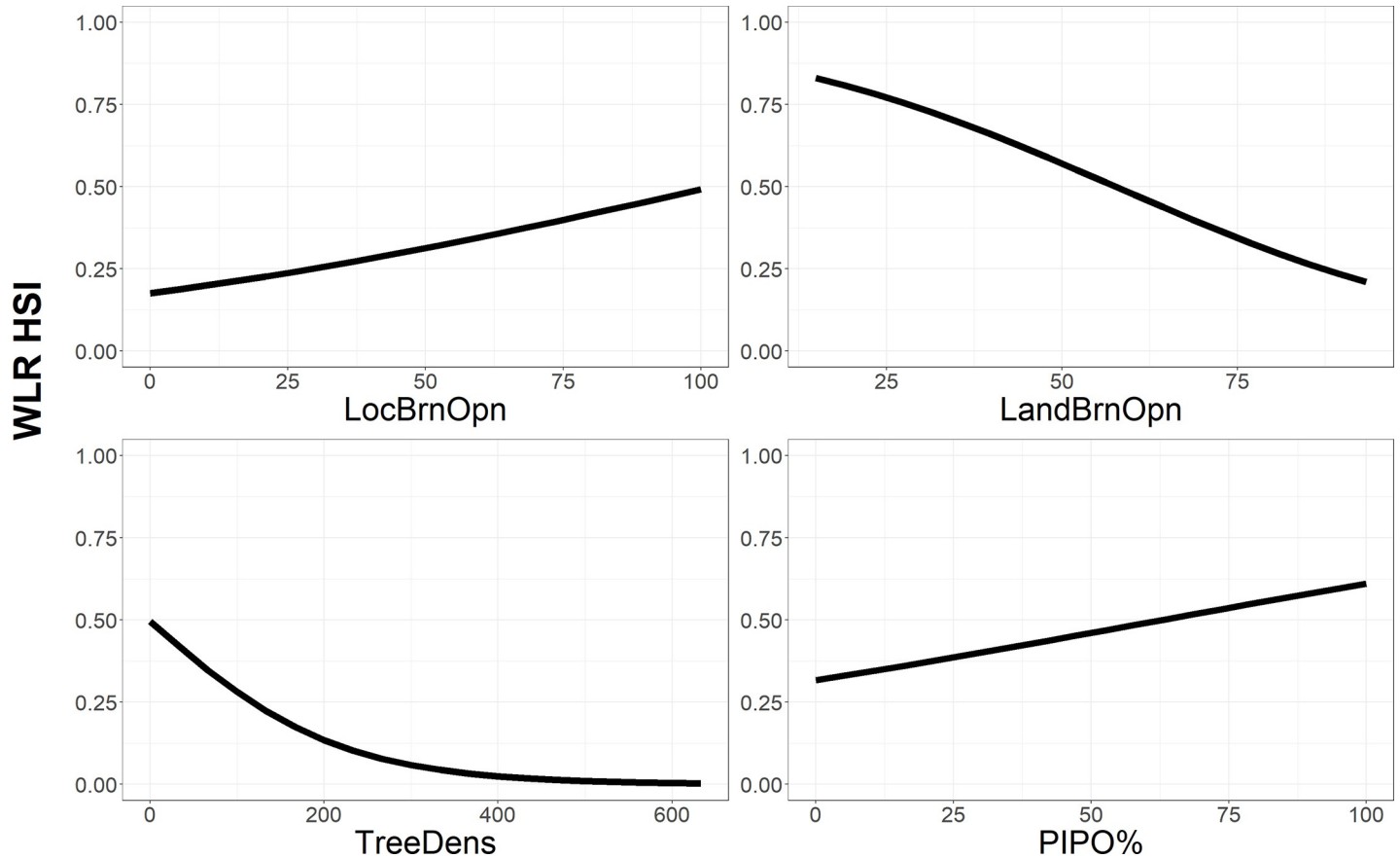

**Fig 3. Weighted logistic regression (WLR) HSI relationships with underlying covariates.** Covariates are local- and landscape-scale percent area burned or open (LocBrnOpn, LandBrnOpn), density of live trees (TreeDens), and percent ponderosa pine (PIPO%). Complete descriptions are in Table 2. The model represented pooled data across Toolbox and Canyon Creek wildfire locations (Oregon).

Canyon Creek, hatched-nest densities were higher overall and differences among suitability classes were less pronounced than at Toolbox. Nevertheless, hatched-nest densities consistently increased from low to moderate and from moderate to high suitability classes at both locations (Table 8).

## Discussion

We found limited transferability of HSI models for nesting white-headed woodpecker in burned forests. Accordingly, we met our objectives within a range of conditions represented

**Table 6. Parameter estimates (and standard errors) for top AIC$_c$-ranked weighted logistic regression habitat suitability index (HSI) models for nesting white-headed woodpeckers in burned forest.** Models were developed with data from Toolbox, Canyon Creek, or both locations combined. Estimates and standard errors describe logit-linear relationships with HSI. Complete covariate names and descriptions are in Table 2.

| Parameter | Developed at: | | |
|---|---|---|---|
| | **Toolbox** | **Canyon Creek** | **Toolbox & Canyon Creek** |
| Intercept | 0.289 (1.661) | 3.691 (1.235) | 0.748 (0.987) |
| LocBrnOpn | 0.037 (0.02) | -- | 0.015 (0.01) |
| LandBrnOpn | -0.058 (0.021) | -0.052 (0.018) | -0.037 (0.012) |
| TreeDens | -0.052 (0.02) | -0.008 (0.003) | -0.009 (0.005) |
| PIPO% | 0.021 (0.012) | -- | 0.012 (0.006) |

**Table 7. AUC scores (with bootstrapped 95% CIs) indicating discrimination accuracy of nest from non-nest sites for white-headed woodpecker in burned forest.** Models were developed at Toolbox and Canyon Creek wildfire locations, and evaluated at both development locations and the Barry Point location. Maxent models were developed with remotely sensed data, and weighted logistic regression models with both remotely sensed and field collected data. AUCs with 95% CIs overlapping 0.5 indicated poor discrimination accuracy.

| Model type | Applied at: | Developed at: | | |
|---|---|---|---|---|
| | | **Toolbox** | **Canyon Creek** | **Toolbox & Canyon Creek** |
| Maxent | Toolbox | 0.76(0.68,0.85) | 0.72(0.62,0.81)[a] | 0.72(0.63,0.81) |
| | Canyon Creek | 0.61(0.52,0.7)[a] | 0.64(0.54,0.73) | 0.62(0.53,0.71) |
| | Barry Point | 0.56(0.37,0.76)[a] | 0.53(0.34,0.72)[a] | 0.57(0.38,0.75)[a] |
| WLR | Toolbox | 0.81(0.74,0.89) | 0.62(0.52,0.72)[a] | 0.76(0.67,0.85) |
| | Canyon Creek | 0.66(0.57,0.75)[a] | 0.71(0.62,0.79) | 0.69(0.61,0.78) |
| | Barry Point | 0.57(0.38,0.75)[a] | 0.55(0.36,0.74)[a] | 0.57(0.38,0.76)[a] |

[a]AUC scores outside where models were developed are of particular interest for assessing limitations to predictive performance and model transferability.

by Toolbox and Canyon Creek locations where models effectively predicted nesting distributions (indicating consistency in habitat relationships). We therefore expect models provided here to be informative for managing post-fire forests within but not necessarily outside this range of conditions (e.g., not at Barry Point). Both Maxent and WLR models showed similar transferability, suggesting both can be informative, the former for mapping suitable habitat to inform conservation planning and the latter to inform post-fire silviculture prescriptions. Hatched-nest densities can facilitate interpretation of HSIs and HSI-based suitability classes and evaluate implications of alternative management scenarios or prescriptions for nesting populations.

The conditions most consistently identified as suitable by models here were canopy mosaics or edges, wherein nest placement favored burned or open-canopy sites adjacent to less-burned and relatively closed-canopy sites. In burned and unburned forests, white-headed woodpeckers generally favor relatively open canopies for nest placement within home ranges that include some closed-canopy forests thought to provide foraging habitat [27, 34, 35]. Modeled

**Table 8. Density of hatched nests in suitability classes defined by HSI thresholds based on Maxent and weighted logistic regression (WLR) models (Maxent thresholds = 0.34, 0.6; WLR thresholds = 0.3, 0.53).** Models were developed with data on nesting white-headed woodpeckers from Toolbox and Canyon Creek burned forest locations (Oregon). 95% CLs (values in parentheses) are bootstrapped with 600 m cells as sampling units. Percent nests is the expected percent of hatched nests assuming equal area sampling across suitability classes. Area surveyed was calculated as the proportion of reference sites (available for Maxent, non-nest for WLR) in each suitability class multiplied by the total area surveyed at each location.

| Model | Location | Quantity | Habitat suitability (HSI) class | | |
|---|---|---|---|---|---|
| | | | **Low** | **Moderate** | **High** |
| Maxent (remotely sensed) | Toolbox | Density | 0.07 (0.01,0.13) | 0.38 (0.22,0.54) | 0.98 (0.48,1.54) |
| | | Percent nests | 5 (1,10) | 26 (16,43) | 69 (50,80) |
| | | Area surveyed (ha) | 7,119.4 | 6,379.9 | 1,530 |
| | Canyon Creek | Density | 0.16 (0.06,0.3) | 0.77 (0.49,1.06) | 1.51 (0.67,2.5) |
| | | Percent nests | 7 (2,14) | 31 (19,50) | 62 (41,76) |
| | | Area surveyed (ha) | 4,902.5 | 3,510.2 | 661 |
| WLR (remotely sensed & field collected) | Toolbox | Density | 0 (0,0) | 0.14 (0.06,0.26) | 0.73 (0.37,1.34) |
| | | Percent nests | 0 (0,0) | 16 (7,32) | 84 (68,93) |
| | | Area surveyed (ha) | 2,188 | 7,019.8 | 3,008.5 |
| | Canyon Creek | Density | 0.19 (0.04,0.43) | 0.29 (0.14,0.5) | 1.02 (0.68,1.49) |
| | | Percent nests | 13 (3,26) | 19 (9,33) | 68 (52,82) |
| | | Area surveyed (ha) | 2,113.8 | 4,124.4 | 2,835.5 |

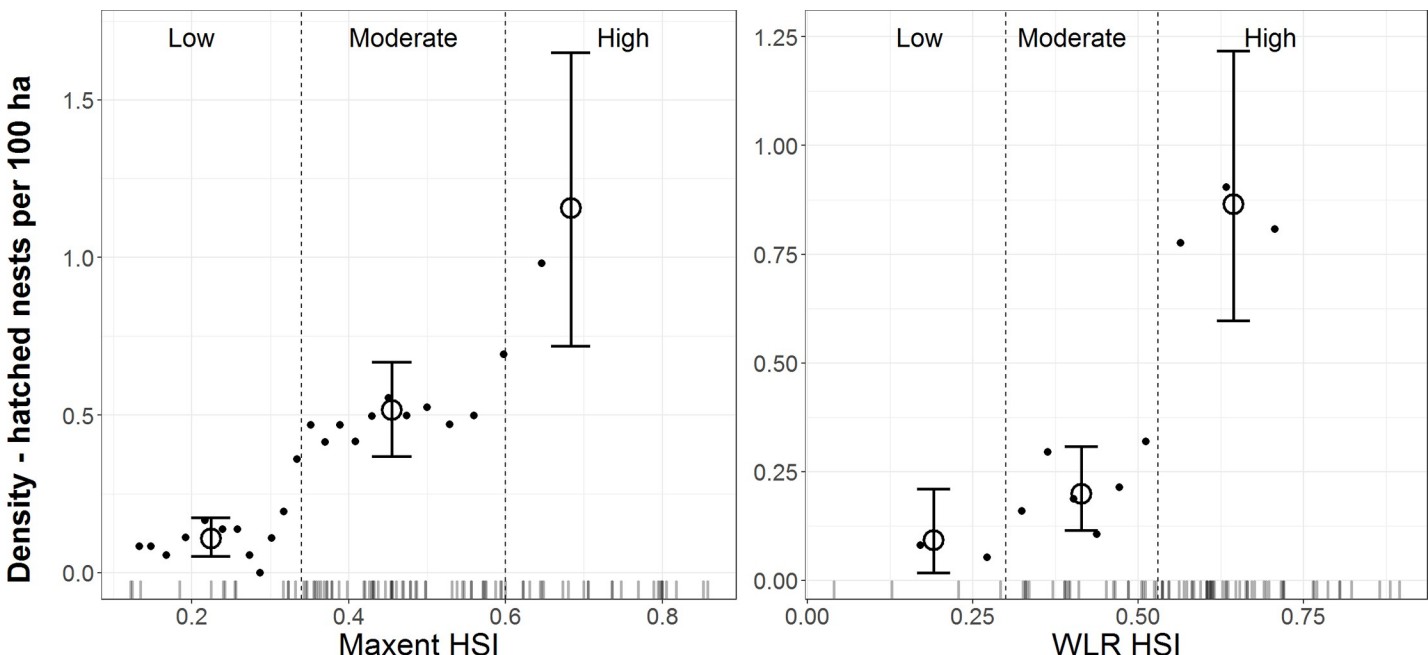

**Fig 4. Densities of hatched nests for white-headed woodpeckers along habitat suitability index (HSI) gradients in burned forests.** Maxent HSIs quantify relationships with remotely sensed environmental variables only, whereas weighted logistic regression (WLR) HSIs also quantify relationships with field-collected variables. Low, moderate, and high suitability classes are differentiated by two HSI thresholds selected at natural breaks in densities for equal-area moving window bins (small circles) and in the distribution of nest site HSIs (rug bars). Large circles and error bars are density estimates and bootstrapped 95% CIs, respectively, for habitat suitability classes.

relationships with LocBrnOpn (positive), TreeDens (negative), and LandBrnOpn (negative) were consistent with this general pattern. Models also described positive relationships with ponderosa pine at various scales (LandPIPO, PIPO%) and a negative relationship with topographic slope, which are consistent with current knowledge [27, 33–35] but were less consistent across study locations. Descriptive statistics for Barry Point sites were also consistent with some of these relationships (e.g., see values for LandPIPO and TreeDens; Table 3). Nevertheless, the combination of covariate relationships quantified at Toolbox, Canyon Creek, or both locations did not readily discriminate nest from non-nest sites at Barry Point. Taken together, these results suggest this suite of features can be informative for quantifying nesting habitat in burned forests with some generality, but not everywhere.

Previously, Wightman et al. [27] developed a Mahalanobis $D^2$ model [57] using nest sites from the Toolbox Fire in central Oregon to map habitat suitability for nesting white-headed woodpeckers. Their model relied on remotely sensed metrics of burn severity, pre-fire canopy cover, and the interspersion-juxtaposition of different forest patches. These metrics were ecologically relevant, but independent data from other wildfires were not available for evaluating predictive accuracy [27]. Our study represents a refinement of Wightman et al.'s [27] work analogous to those made for white-headed woodpecker nest habitat models implemented in unburned forests [35]. Additionally, the size and density of snags and trees are not well represented by remotely sensed data, so including field measurements adds important information for quantifying nesting habitat relevant to management (e.g., salvage logging) prescriptions.

## Limitations to model transferability

Differences in habitat availability and consequent nest site selection likely caused limited transferability of models to Barry Point from other wildfire locations. Although forests at the three study locations all represented lower elevation dry conifer forests, forest coverage and composition varied among locations. Ponderosa pine-dominated forest was least extensive at Barry Point, and forest patches were interspersed more with non-forest openings composed primarily of sagebrush (*Artemisia tridentata*), mountain mahogany (*Cercocarpus ledifolius*), and juniper (*Juniperus occidentalis*). These open-canopy sites were present prior to wildfire, so assuming availability of snags suitable for nesting at these sites, white-headed woodpeckers at Barry Point may not have depended as heavily on wildfire to generate canopy openings desirable for nesting. Ponderosa pine-dominated forest was less extensive at Barry Point and nearly 40% of nests were located in juniper trees situated in open-canopied forests (*n* = 19 nests). Relatively large ponderosa pine (DBH ≥ 25 cm) provide nesting and foraging substrates, especially high quality foraging resources [e.g., 58]. In contrast with their counterparts at Toolbox and Canyon Creek, white-headed woodpeckers at Barry Point may have focused selection less towards canopy mosaics and instead towards ponderosa pine-dominated forests, where they likely found trees desirable for foraging. Nesting relationships with size and configuration of forest patches and canopy openings and comparison of such metrics across wildfire locations could help evaluate these potential explanations for our results.

Even at Toolbox and Canyon Creek locations, we observed differences that could impose limits on model applicability and generality. Toolbox sites were characterized by more extensive coverage of ponderosa pine-dominated forest and less topographic slope than at Canyon Creek. These differences may reflect variation in selectivity for these features between wildfire locations. Models may need to allow relationships with slope and percent ponderosa pine to be modulated by habitat availability to correctly inform predictions [sensu 23].

Salvage logging could also influence model transferability. During preliminary analyses, salvage logging covariates reduced predictive performance (AUC ≈ 0.5). This poor predictive performance did not necessarily indicate a lack of relationships with salvage logging but rather that such relationships were too inconsistent to reliably inform prediction across study locations. Descriptive statistics did in fact suggest somewhat contradictory relationships with logging at Canyon Creek versus Toolbox (Table 3). We therefore excluded salvage logging covariates from models here to maintain focus on developing predictive models. Studies specifically examining salvage logging effects could complement predictive habitat models to inform forest management. Selective logging may sometimes improve habitat suitability by creating canopy openings desirable for nest placement, but logging effects will likely vary among wildfire locations with different logging prescriptions. Data collected over a range of logging prescriptions and pre-logging environmental conditions are needed to quantify generally applicable relationships with logging. Salvage logging may alter interpretation of variables based on remotely sensed pre-fire canopy data, potentially necessitating greater reliance on field-collected data in heavily logged areas. Salvage logging levels (extent and intensity) represented at our study locations did not compromise the value of remotely sensed data for characterizing nesting habitat and predicting nesting distributions for white-headed woodpeckers.

Model transferability can vary with modeling technique and data quality [59–61]. Previous study shows improvements to transferability when including field-collected data [32]. Here, WLR models not only included field-collected data, but were also developed with use–non-use (nest–non-nest) data, which are generally expected to provide higher quality information than data without non-use (i.e., absence) data [15, 49]. Nevertheless, we found comparable transferability with Maxent models informed only by remotely sensed and use-availability (i.e.,

presence-background) data. Factors such as modeling technique or sample size may compensate somewhat (but probably not entirely) for reduced information quality where field-collected and non-use data are unavailable.

Following our primary objective, we took an approach often represented in machine learning studies wherein we exhaustively considered potential combinations of candidate covariates and evaluated predictive performance to verify the utility of the resulting model [see also 59–63]. Selection from a narrower set of candidate models representing *a priori* hypotheses [described in 51] may be more appropriate for research investigating mechanisms underlying observed habitat relationships, which would complement and inform development of predictive models.

## Model application guidelines

We suggest using the pooled Maxent model to map habitat to inform selection of habitat reserves for white-headed woodpeckers and/or salvage logging units following wildfire, whereas the pooled WLR model (along with descriptive statistics of field-collected data) provides finer resolution information relevant to designing management prescriptions. We expect most applications will require categorization of habitat (e.g., as low, moderate, or high suitability). Hatched-nest densities in relation to HSI categories can facilitate their application to inform management and compare alternative management scenarios to meet particular population targets. Management plans could aim to maximize retention of moderate and high suitability habitats classified by the pooled Maxent model. Silviculture prescriptions could target conditions associated with high suitability habitat classified by the WLR model. Conditions within a 1-km radius neighborhood of nest sites informed models, so buffering habitat reserves and treatments would be needed to provide sufficient foraging habitat and maintain nest habitat suitability as described by these models. We developed a toolset to apply HSI models for disturbance-associated woodpeckers (including models presented here) within a GIS framework, along with a manual demonstrating potential applications [64].

Models provided here do not comprehensively quantify all habitat features required for nesting. Given the potential importance of ponderosa pine-dominated forest and topographic slope suggested at individual wildfire locations, we suggest restricting Maxent HSI application to a range of conditions corresponding with where we surveyed (i.e., 1-km radius neighborhood coverage of ponderosa pine-dominated forest [LandPIPO] $\geq$ 40% and Slope $\leq$ 40%). Descriptive statistics here (Table 3) combined with other studies [27, 33] suggest logging prescriptions that retain relatively large decayed snags would benefit nesting white-headed woodpeckers (Table 3). Although models suggest local snag densities were less important than other habitat features for selecting nest sites, intensive salvage logging could reduce habitat suitability by limiting snag-related resources. Experimental study examining population response to a range of logging prescriptions could complement HSI models for informing post-fire forest management. Finally, habitat selection does not always optimize fitness [65, 66], so habitat-fitness relationships would complement HSI models to inform habitat management [e.g., 27, 34].

## Towards more predictive models

Studying mechanisms underlying observed habitat relationships would further inform predictive modeling. Ponderosa pine trees provide multiple foraging opportunities (cones provide seeds and invertebrates; and bark, needles, and pine sap provide insects) and desirable substrate for nest cavity excavation [33, 58, 67–69]. The association of white-headed woodpeckers with canopy mosaics represents a more recent discovery and is not fully understood. Canopy

openings generated by wildfire could provide refuge from nest predators [e.g., red squirrel [*Tamiasciurus hudsonicus*]; [70] and opportunities for aerial insectivory [5, 44, 71]. In contrast, adjacent unburned closed-canopy forests are thought to provide critical opportunities for foraging on live ponderosa pine trees. Following wildfire, white-headed woodpeckers could also find foraging opportunities in high densities of recently burned snags. These hypotheses are largely untested, and their importance has implications on the level, scale, and character of canopy mosaics that optimize habitat suitability, as well as behavioral plasticity in the use of mosaic habitats. Improved understanding of resources provided by canopy mosaics would further inform how to quantify mosaics in predictive models. Additional tree-level data could help refine models that include field-measured covariates to inform management prescriptions (e.g., nesting use of juniper trees and metrics of snag decay). Following previous work and our understanding of underlying mechanisms, we chose to estimate the expected association with canopy mosaics by estimating relationships at different scales within the same model [22]. Future research could also consider quadratic relationships with canopy cover and burn severity as an alternative or additional approach for quantifying these relationships.

Contemporary best practices include model averaging to account for model-selection uncertainty when generating model-based predictions [72]. Practitioners could average across WLRs presented here to infer potential management effects on habitat suitability. For transparency and interpretability towards informing the design of management plans and prescriptions, however, we opted to select a single best model. Although our selected model represents the best balance of information and parsimony given available data, our conception of which covariates are necessary and sufficient for prediction could evolve with additional sampling and study of underlying mechanisms. Nevertheless, we expect to retain major habitat components of canopy mosaics and ponderosa pine even with further model refinements [e.g., 35].

## Broader implications

Management of dry conifer forests is currently focused on restoring and maintaining forest conditions with which white-headed woodpeckers are closely associated and that have been disrupted by human activities [27, 33, 34, 35]. Because of their association with these conditions, white-headed woodpeckers have now been adopted as a focal species for assessing the effect and efficacy of forest management treatments and strategies [e.g., 73, 74]. Some evidence suggests burned forests may represent essential habitat for white-headed woodpecker population persistence [27, 34]. The species consequently draws particular attention from managers seeking to balance socioeconomic demands for timber and public safety with habitat conservation when planning salvage logging. Models can generally inform post-fire forest management that targets habitat for white-headed woodpeckers in the East Cascades and Blue Mountains (northwestern U.S.A.), but additional data are needed in other portions of the species range (e.g., Modoc Plateau, represented here by Barry Point, and North Cascades).

## Supporting information

**S1 Appendix.**
(DOCX)

**S2 Appendix.**
(DOCX)

**S1 Data.**
(DOCX)

**S1 File.**
(ZIP)

## Acknowledgments

Field crew supervisors that provided oversight of data collection included C. Forristal, D. Hopkins, K. Nicolato, and M. Davies. Logistical support was provided by B. Yost, C. Reames, A. Unthank, and L. Stokes. We are grateful to all field assistants. We thank L.S. Baggett and A. Johnston for thoughtful reviews of earlier drafts.

## Author Contributions

**Conceptualization:** Quresh S. Latif, Victoria A. Saab.

**Data curation:** Quresh S. Latif, Jonathan G. Dudley.

**Formal analysis:** Quresh S. Latif.

**Funding acquisition:** Victoria A. Saab, Amy Markus, Kim Mellen-McLean.

**Investigation:** Victoria A. Saab, Jonathan G. Dudley.

**Methodology:** Quresh S. Latif, Victoria A. Saab, Jonathan G. Dudley, Amy Markus, Kim Mellen-McLean.

**Project administration:** Victoria A. Saab, Jonathan G. Dudley, Amy Markus, Kim Mellen-McLean.

**Resources:** Amy Markus, Kim Mellen-McLean.

**Supervision:** Victoria A. Saab, Amy Markus.

**Writing – original draft:** Quresh S. Latif.

**Writing – review & editing:** Victoria A. Saab, Jonathan G. Dudley.

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
