## [Decision Letter · Decision Letter 0]

4 Feb 2020

PONE-D-19-34065

Development and evaluation of habitat suitability models for nesting white-headed woodpecker in burned forest

PLOS ONE

Dear Dr. Latif,

Thank you for submitting your manuscript to PLOS ONE. After careful consideration, we feel that it has merit but does not fully meet PLOS ONE’s publication criteria as it currently stands. Therefore, we invite you to submit a revised version of the manuscript that addresses the points raised during the review process.

This paper provides an important contribution to the challenge of balancing the needs of native species, such as the white-headed woodpecker, and forest management objectives.  As both of the reviewers point out, the paper is well written but could be improved by some revisions.  Many of the suggestions will increase the utility of the paper and provide additional desirable details.  I will highlight a few of these but I encourage you to read through the reviewers comments carefully as they make very excellent suggestions for ways to improve the manuscript. 

One key aspect of this study is the development of HIS models for woodpecker nesting habitat following fire management applied to several novel locations (e.g., transferability).  As the results suggest, it is critical to evaluate whether models can be applied more generally or are highly dependent on local context.  This is a strength of the paper that is not highlighted as well as it could be.  For example, there is no direct comparison of habitat suitability with only remote sensing data only versus with only field-collected data, although only remote sensing data is compared to the combined field and remote sensing data. This omission makes it challenging to assess the value of the data at different scales.  The white-headed woodpecker seems like an excellent model to test this but there may need to be some additional information provided about their ecology to better frame this discussion as Reviewer 1 points out. 

 I agree with both of the reviewers that the analyses are thorough and, for the most part, well described.  However, there are a few issues highlighted by both reviewers worth noting.  In particular, there are clearly far more models that you describe than are listed in the paper.  It would be very valuable for the reader to be able to evaluate the larger set of candidate models.  As reviewer 2 points out, there is a revised understanding of AIC in the literature that would encourage going beyond the models within 2 AICc units of the top model.  It would also be valuable to provide additional information in Table 6 where you compare the top set of models for each location (e.g., Akaike weights).  Reviewer 2 points out that descriptive statistics are mentioned but not described or displayed.  Some other details are mentioned about salvage logging and I agree with reviewer 1 that if that is to be a focus of the paper than more details are needed in both the methods and the results to provide the support for the discussion.  Otherwise, this aspect may need to be de-emphasized.  Both reviewers provide detailed suggestions on the figures and tables that would strengthen the paper (e.g., combining Tables 2 and 3).  Also, more detail is needed in your description of the models, their parameters, and the assumptions for each approach (e.g., reviewer 1 comments specifically on weighted linear regression, reviewer 2 makes suggestions about model evaluation).  Additionally, a different set of reference sites are used for each of the models.  It is important to explain why these changes are made and what potential implications they have on the results.  I agree with reviewer 1 that the hatched-nest analysis is not well described, which undermines its purported value.  Understandably there are a lot of these details that may need to be added to a supplement but they would strengthen the usability of your approach by others.

The goal of the study is to inform post-fire forest management by identifying suitable habitat for fire-associate species.  You mention the evaluation of management scenarios but this is not demonstrated in the results unless you count the different sites as different scenarios; it would be very valuable to see some example scenarios modeled.  In addition, to explore the issue of transferability the models could be refined with inclusion of a measure of habitat availability, as you provide this a possible explanation for the differences that you found.  Reviewer 1 makes some very constructive suggestions to improve the discussion and better tie the paper together and reviewer 2 highlights some of the elements that need clarification.

Overall, this paper will be a strong contribution to the literature but it needs some revision.  The paper does a very nice job of highlighting two different approaches to modeling habitat suitability while highlighting some of their strengths and weaknesses.  The conclusions are important and seem supported by the data, although some improvement in the presentation will strengthen this connection.

We would appreciate receiving your revised manuscript by Mar 20 2020 11:59PM. To enhance the reproducibility of your results, we recommend that if applicable you deposit your laboratory protocols in protocols.io, where a protocol can be assigned its own identifier (DOI) such that it can be cited independently in the future. For instructions see: http://journals.plos.org/plosone/s/submission-guidelines#loc-laboratory-protocols

We look forward to receiving your revised manuscript.

Kind regards,

Karen Root, Ph.D.

Academic Editor

PLOS ONE

Journal Requirements:

2. Our internal editors have looked over your manuscript and determined that it is within the scope of our Biodiversity Conservation Call for Papers. This collection of papers is headed by a team of Guest Editors for PLOS ONE (https://collections.plos.org/s/biodiversity). The Collection will encompass a diverse range of research articles on biodiversity conservation, including advances in conservation solutions of species or ecosystems. Additional information can be found on our announcement page: https://collections.plos.org/s/biodiversity

If you would like your manuscript to be considered for this collection, please let us know in your cover letter and we will ensure that your paper is treated as if you were responding to this call. If you would prefer to remove your manuscript from collection consideration, please specify this in the cover letter.

5. Please upload a copy of S1 Appendix and S2 Data which you refer to in your text on page 35.

Reviewers' comments:

Reviewer's Responses to Questions

**Comments to the Author**

1. Is the manuscript technically sound, and do the data support the conclusions?

Reviewer #1: Yes

Reviewer #2: Yes

2. Has the statistical analysis been performed appropriately and rigorously? 

Reviewer #1: Yes

Reviewer #2: Yes

3. Have the authors made all data underlying the findings in their manuscript fully available?

Reviewer #1: Yes

Reviewer #2: No

4. Is the manuscript presented in an intelligible fashion and written in standard English?

Reviewer #1: Yes

Reviewer #2: Yes

5. Review Comments to the Author

Reviewer #1: Excellent and well-written paper. Thanks for the opportunity to review it. This paper will add greatly to our knowledge of white-headed woodpecker use of post-fire landscapes. It will also be useful for forest managers who are faced with increasing fire severity and frequency in the northwestern U.S., where this species is of conservation concern. As the authors point out, managers in these areas are tasked with difficult decisions – salvage logging is done to meet timber targets but can be detrimental to this and other at-risk cavity nesters. Most of my comments were minor or relating to semantics – as well as trying to improve the readability for individuals not familiar with white-headed woodpecker nesting ecology. I commend the authors on a great manuscript and hope this will be published soon, so it can be used by managers.

My biggest concern is that the paper starts out with an emphasis on salvage logging, but in the results and discussion the salvage log results are glossed over because they did not improve the predictive value of the model. To fix this, the authors could re-write the abstract and introduction to remove the focus on salvage logging. Managers need information on salvage logging however, so I’d prefer instead that the authors provide some of the models that did include salvage logging, and/or do more with the descriptive statistics on salvage logging. At the very least, the authors could start the discussion with a paragraph that describes the effects of salvage logging on this species. Even if the paragraph simply states/clarifies that the levels of salvage logging examined in this study had no effect on habitat suitability for white-headed woodpecker.

Specific comments:

Line 61 – suggest changing wording to “managers must strive to identify suitable habitat….”

Line 72 – suggest removing “project”

Line 85-86 – many things can cause geographic variation in model applicability. Suggest adding the following to the beginning of this sentence “Many factors can cause geographic variation in model applicability, however, such as biotic interactions, local adaptation and behavior rules….”

Line 98-99 – soften “require”; suggest replacing with the word “use”, or something similar; not enough is known of this species for such strong language, plus it does not necessarily account for variation in habitat use range-wide

Line 98 – sentence is confusing because it starts out that “white-headed woodpeckers nest in….unburned forested landscapes….. maintained by mixed severity fire”. Do you mean to imply that historically the unburned forests were maintained by mixed severity fire? Otherwise unburned forests being maintained by mixed severity fire seems incongruous. Also note in southern portion of this species range conditions appear different – white-headed woodpecker may not require areas maintained by mixed severity fire in southern CA, for example. Too little research has been done in such locales to be sure either way.

Line 101 – suggest removing “critical”

Line 103 – after ponderosa pine, add “in the northern portion of their range”

Line 104 – remove “suggesting possible source-sink dynamics” or add a sentence to clarify. As written, it is confusing to the reader.

Line 108 – Use of “further” here seems out-of-place. Suggest deleting.

Line 110 – the objectives 1 and 3 are difficult to understand as written. From reading the paper, objective 1 seems to be to develop models of nesting habitat suitability using GIS covariates and field covariates at 3 fires that had some level of salvage logging. Objective 2 is understandable as written. As written, objective 3 seems to be the same as objective 1.

Line 113 – what is meant by the phrase “considering our success in achieving these objectives”? Suggest deleting and start the sentence with “We also evaluated our ability to quantify….”

Table 1 – I appreciate the detailed info in this table, as well as the detailed comments indicated by subscripts.

Line 153 – Use of “affected” is confusing. Suggest stating instead that “Portions of the toolbox and canyon creek fires were salvage logged.” Or clarify early on in this paragraph what is meant by “affected”.

Line 153 – Photos of different fires/study areas – even black-and-white – would be nice to add for the reader.

Line 195 – 35 m is apparently the minimum distance What was the mean and range of distance the non-nest measurements were from the nests? See comment relating to line 300.

Table 2 – Is LandPIPO pre-fire live or post-fire live ponderosa pine?

Line 243 – 25 cm dbh does not seem large to me. Also, in Table 3 large snags are those >50 cm. Suggest deleting the word “large” and simply state that white-headed woodpeckers nest mostly in snags >= 25 cm. I suggest this change be implemented throughout the document. Avoid vague qualifiers “small” or “large” and use numbers (e.g., snags 25-25 cm, > 50 cm, etc).

Table 3, also line 380 – Decay is so vague given the guidelines (yes/no whether ‘nest or center tree showing signs of decay’), plus it wasn’t recorded at one site, so I think it should be removed from the document. It will only confuse the results plus the authors do a nice job on line 252 describing why decay was not modeled. Remove it from the document and simply state that measuring decay is difficult and we did not measure it in this study.

Table 3 – It might be nice to merge tables 2 and 3 so the reader has all the model covariates in one table. I also suggest separating out the covariates that were not included in models. Right now they are indicated with small subscripts. You could have two subheadings: (1) measured covariates that were included in models, and (2) measured covariates that were not included in models.

Line 300 – The weighted logistic model assumes use/nonuse data, in other words? (e.g., Nad’o and Kanuch 2018. Plos one 13:e0200742; rather than use/availability data, like Maxent..) Perhaps this is discussed in detail later, but further consideration of the violations of this assumption are needed here, especially given this is a territorial species and ‘non-use’ sites were apparently as close as 35 m to nests (line 195), which is within an area likely defended by territorial pairs (given that white-headed woodpecker inter-nest/nearest neighbor distances in other studies are typically >100 m). If this species is territorial around nests, it seems it would have been better to buffer around nests before drawing a sample of random points to be used in a use/nonuse design. What if one white-headed woodpecker had a nest in a ‘suitable’ habitat pixel, but it was surrounded by say 2-5 ha of other suitable pixels. Those 2-5 ha of suitable pixels could not have been used by another nesting white-headed woodpecker because of territoriality, and thus may have been incorrectly classified as unsuitable habitat in this design; this may bias the HSIs. Temporal aspects could also be considered – a pair may have used one of the unused pixels the year after you finished the study, even though the suitability for nesting did not change among years. It seems a use/availability SDM is better over all for this study (which is true of most wildlife studies, really). If WLRs are kept in the study to enable analysis of the field-collected variables, please be clear in the text here that you may have violated some assumptions – and be transparent about those assumptions so readers less familiar with SDMs can understand that those results may be a bit more questionable.

Line 303 – I get what the authors are trying to say with this sentence (WLR HSIs primarily quantified relative suitability…) but this argument would need a fair amount of justification – seems more than the scope of this paper. Please consider deleting this sentence.

Line 328 – by “informative”, do you mean transferable?

Line 337 – Why was the proportion of available sites was set to match the proportion of nest sites for the pooled maxent model?

Line 342, and throughout – Especially here in this section describing pooling of study sites, the word ‘location’ gets confusing – just because it has so many meanings. Can the authors use the word “study area” or ’study site’ or ‘fire complex’ consistently throughout the manuscript (and in this section) instead of the word location.

Line 343 – I do not understand this sentence, and thus I do not understand what the ‘hatched’ nests are being used to validate. This paragraph is difficult to follow. (line 343-350) After reading the results it became clear to me that the hatched nests are being used to identify thresholds for low, moderate, and high suitability but I think that has limited value and I suggest deleting the thresholds from the manuscript. See comment for line 481.

Table 4. It is nice to have a table showing the mean values for each covariate.

Line 404 - This statement is unclear to me “supporting a scale-specific tradeoff with burn severity and canopy openness”. Suggest deleting it here, and moving any pertinent argument for or against this concept to the discussion.

Line 462 – Could the authors also present a table with sensitivity and specificity so readers can see in what direction these models performed best (I mean, were the models better at predicting true positives or true negatives?).

Tables 7 and 8. Write out the fire names in the column headings. There are enough acronyms in this table that the reader has to look up without abbreviating the fire names too.

Table 8 - Similarly, suggest removing RS and RS&FC from the first column. Those items are described in the table heading, as well as in the methods, and make the column more confusing than it needs to be. It might be ideal to separate out AUC values for the models transferred to other locations (I mean place those with the subscript “a” in a different table, or in a subheading of this table).

Line 481 – Building off my comment associated with line 343, I do not see the value of the “hatched-nest” analysis. I do think in the methods line 343 the authors could do a better job describing what this analysis is aimed to accomplish. It seems to be used to establish thresholds, but why are thresholds needed? Thresholds are tricky, and are not needed for applying habitat models. They can be quite sensitive – I mean that small changes in the value of thresholds can lead to large changes in the amount of area classified as suitable. Moreover, there are more established ways of assigning thresholds from SDMs like Maxent. I think this analysis should be dropped, and the suitability of habitat from the models be assessed based on the numerical value of the HSIs at given locations. If the authors do choose to keep this analysis, it may help to add it as an objective to the introduction, and please make the methods around line 343 more understandable.

Line 534 – This is well-written.

Line 556 – The species requires snags when breeding so this statement would only hold true if there were snags in the open sites at Barry Point - before the fire.

Line 560 – similar to previous comments, I’d remove the qualifier ‘large’ (and small!) as much as possible. It only adds to confusion among managers about how to provide habitat for this species, which research has shown nests in areas with heavy timber production – thus relatively ‘small’ ponderosa pine trees can also provide nesting and foraging substrates. Best to just use numbers, rather than vague verbiage like large and small.

Line 569 – Use of ‘modulate’ is unclear here.

Line 611 – Poor word choice with ‘favor’. I think the authors mean to imply that prescriptions that protect large decayed snags would benefit white-headed woodpecker. The use of the word favor here at first made me think the authors were suggesting prescriptions should harvest large snags.

Line 616 – This is well written.

Line 625 – How does this sentence relate to white-headed woodpecker? For one thing, nest predators are largely unknown for this species and there is no point in perpetuating theories that are not well-supported by data. Second, the literature indicates that aerial insectivory is not a common foraging method for this species.

General comment – the word mosaic seems a bit overused and is not common parlance. My main reason for commenting is it may confuse readers. Could the authors use the word ‘patchy’?

Line 620 – in general, this section does not add much to the paper. I suggest deleting it.

General comment for the discussion – Two paragraphs would help tie this paper together more nicely: (1) First, start the discussion with a paragraph on the effects of salvage logging, because that is how the abstract starts (and salvage logging is mentioned early in the introduction also). Readers who want to skip the methods and results could then zoom down to the discussion, and the first paragraph would then inform them of the overall effects of salvage logging on this woodpecker species. Even if salvage logging did not impact the habitat suitability, it would still be valuable to start the discussion with a statement to that effect, along with a brief review of the salvage log prescription that were analyzed in this study. (2) add a management implications section and describe implications of results pertaining to salvage logging.

Reviewer #2: The authors have submitted a well-written manuscript that reports on the use of Maxent to make predictions about habitat suitability for White-headed Woodpeckers at burns in Oregon, USA. The basic approach was to compare nests vs non-nest sites at three burns. Two of the sites appeared to have better predictive ability but the third site differed, making any transferability inappropriate.

I really liked the manuscript and appreciate the depth of analyses that went into the patterns reported in the manuscript. Coupled with the extensive field data, this manuscript represents a lot of effort! My comments are mainly editorial in nature.

The main issue I have is with the dumbing down of quantitative analyses. Rather than telling the reader that previous analyses didn’t show any difference so we only present the simplified version, the authors should provide all of the information and let the reader decide for themselves. Likewise, if the authors are going to discuss analyses that were conducted, they need to present those analyses. And, finally, if the authors are going to rank candidate models, show the reader ALL of the models rathe than filtering it down to just the delta AICc < 2. I find this particularly annoying.

Here are more specific comments on this issue:

130 More details need to be provided on this “preliminary analysis”. What is meant by “substantially different relationships”?

288-290 Is this statement about verifying comparable performance with more complex models documented anywhere? The reference to citation 35 pertains to a 2015 paper comparing nest use HSIs, with presumably an entirely different database. I think this information should be included in an appendix.

311-312 Is this statement about only using first-order linear covariate relationships related to the previous statement about using simpler models? As before, this information should be provided in a supplement / appendix.

366 I did not have access to S2 data so was unable to evaluate or provide feedback. Likewise, at line 851, the authors indicate there is a zipped folder with a git repository. None of this was available when evaluating this manuscript.

414 The authors compare the global to a reduced subset of models. I thought the authors previously said they did not use more complex models. Please clarify. At the very least, the reader needs to know what models were in the global models vs the reduced models if the authors are going to reference global vs reduced in Table 5.

439 The authors present only those candidate models within 2 AICc units from the top model. The full suite of models should be provided in a supplement / appendix. The reader would be interested in knowing how many models were withing 2-3 AICc units. Was there a clear break or was there a continuum of supported models that extended beyond 2 units? This dumbing down of the data presentation can be particularly annoying to readers that want to see the full suite of candidate models.

452 Most modern day information theorists recommend abandoning the delta < 2 mantra. See extensive discussion of this by Burnham et al. (2011). They advocate against that arbitrary cutoff. My recommendation to the authors is to adopt a more modern approach where any of the models with AICc<10 are considered. The obvious solution would be to use model-averaging instead of only relying on the <2 models.

Here is what Burnham et al. (2011) write:

Δ>2 Rule. Some of the early literature suggested that models were poor (relative to the best model), and might be dismissed if they had Δ>2. This arbitrary cutoff rule is now known to be poor, in general. Models where Δ is in the 2–7 range have some support and should rarely be dismissed. Inference can be better based on the model likelihoods, probabilities, and evidence ratios and, in general, based on all the models in the set. From these quantitative measures one can then assign their own value judgment if they wish.

Burnham, K. P., D. R. Anderson, and K. P. Huyvaert. 2011. AIC model selection and multimodel inference in behavioral ecology: Some background, observations, and comparisons. Behavioral Ecology and Sociobiology 65:23-35.

573 Like I have mentioned previously, details about analyses used need to be documented. What are these “descriptive statistics” and where can they be found? Similarly, lines 610-611 discuss “descriptive statistics” about “logging prescriptions” favoring large decayed snags.

Line Edits

20 Insert scientific name for the woodpeckers

35 AUC is not defined; likewise it is not defined in the main text (line 275) at first mention. Instead, it is defined all the way down at 325.

36 Add two decimals to 57.00 so format matches 0.53

90 Define the term “transferability”

127 Insert “(hereafter Canyon Creek Fire;”

134 The sc name was already introduced at line 103.

135 The authors are inconsistent on whether they introduce e.g. or i.e. within parentheses or with merely a comma. This problem occurs throughout.

137 All numbers in the thousands should have a comma for consistency. The authors arbitrarily include the comma or not. See 1,603; 4,347; 4,727; and subscript e for example in Table 1. See also line 358 among other places.

165 Should be “prior to”

168 This is the first mention of DBH so it should be “diameter at breast height (DBH) > 23 cm per hectare”. The authors introduce the abbreviation before the definition.

204 Add extra hyphen “30-m-pixel”

218 The authors need to provide a citation for the statement about the woodpeckers nesting in canopy openings.

266 Typo “diamter”

327 The statement that AUC=1 indicates perfect discrimination needs a citation.

328 This line is the first occurrence of “confidence interval” so it should be abbreviated here (not later in the ms at line 354 as it is currently formatted).

344 This is the first introduction of “hatched nests” so the “(i.e. nests with at least 1 nestling)” information should appear here, not on line 347.

346 The authors indicate that by using hatched nests, detection probability was nearly perfect. How does this approach account for nests that failed before hatching? Does this not bias towards habitat with successful nests (e.g. those areas with less predators, unsuitable nest trees, etc).

352 Two nests were excluded plus any other nests that went undetected, of course.

414 This is the first time the abbreviations TB and CC have been used. They need to be defined before the abbreviation can be introduced like this in the table.

456 At some point, the authors will want to clean up their tables so that the numbers are aligned by decimal point. I appreciate that they may not have done this for the submitted version of their manuscript.

470 Remove hyphen in bootstrapped so it matches the style used throughout the ms.

559 Unclear whether this 40% figure comes from the literature or is based on unpublished data. Either way, the authors need to provide a citation (if published) or a sample size that this percentage is based on if this is unpublished data.

570 Change “correct” to “correctly”

581 Unclear what levels are being referenced here. Levels of what?

594 Are the authors speaking more generally here or are they specifically referring to WHWOs when they mention “habitat reserves”?

598 Insert hyphen “hatched-nest densities”

600 Is this objective of reaching “particular population targets” in play? If so, a citation and / or more background information should be provided here.

610 Unclear what this 40% number represents. How is “dominated or co-dominated by ponderosa pine” defined? Does one just count up all of the trees within a 1-km radius and calculate what proportion of trees are PP? If so, how does one define co-dominated? Some folks are not as up on forestry lingo and need a bit of help understanding the terminology.

617 Are the authors advocating for habitat-fitness data as a prerequisite for constructing HSI models, or just that it would be nice to have? Ambiguous as worded.

641 Citation needed for the statement about restoring and maintaining forests and the disruption by humans.

648 These geographic terms (e.g. East Cascades, Blue Mountains, Modoc Plateau, North Cascades) will mean nothing to a casual reader. Please provide geographic reference to put in context. Oregon? USA?

6. PLOS authors have the option to publish the peer review history of their article (what does this mean?). If published, this will include your full peer review and any attached files.

Reviewer #1: No

Reviewer #2: No

---

## [Author Response · Author response to Decision Letter 0]

20 Mar 2020

We have responded to all reviewer and editor comments on the content of our manuscript in the document entitled "Response to Reviewers" uploaded with this submission.

---

## [Decision Letter · Decision Letter 1]

15 Apr 2020

PONE-D-19-34065R1

Development and evaluation of habitat suitability models for nesting white-headed woodpecker (Dryobates albolarvatus) in burned forest

PLOS ONE

Dear Dr. Latif,

Thank you for submitting your manuscript to PLOS ONE. After careful consideration, we feel that it has merit but does not fully meet PLOS ONE’s publication criteria as it currently stands. Therefore, we invite you to submit a revised version of the manuscript that addresses the points raised during the review process.

This paper provides an important contribution to the challenge of balancing the needs of native species, such as the white-headed woodpecker, and forest management objectives and needs only minor revisions.  I appreciate your thorough attention to the suggestions and questions of the initial reviewers.  As a result of these changes, the paper is much stronger, better written and provides more compelling conclusions.  The revised methods have improved clarity and better highlight the important aspects of the approach utilized.  I appreciate the revision of the analysis to address issues mentioned by the reviewers and the results now better support the conclusions. The discussion provides an excellent overview of the results in a broader context and makes valuable suggestions on future research.  Reviewer #2 has some minor suggestions that you should address in your revisions.  Please, note, though, that Reviewer #3 has highlighted some concerns about the analytical methods leading to their decision to reject.  You will certainly need to address the issue of quadratic effects and provide clarification about why that might not be appropriate or feasible for these data and/or the application intended.  While I agree that it might be worthwhile to include some additional landscape analyses to further explore configuration effects, there are a seemingly endless number of variables you could include in HSI models.  However, the suggested analyses (e.g., fragstats) do not seem necessary if the potential limitations in the conclusions you can make with the current analysis are explicitly acknowledged in the discussion and you think these additions are beyond the scope of the study.  Overall, this paper will make a strong contribution to the field with some additional minor revisions.

We would appreciate receiving your revised manuscript by May 30 2020 11:59PM. To enhance the reproducibility of your results, we recommend that if applicable you deposit your laboratory protocols in protocols.io, where a protocol can be assigned its own identifier (DOI) such that it can be cited independently in the future. For instructions see: http://journals.plos.org/plosone/s/submission-guidelines#loc-laboratory-protocols

We look forward to receiving your revised manuscript.

Kind regards,

Karen Root, Ph.D.

Academic Editor

PLOS ONE

Reviewers' comments:

Reviewer's Responses to Questions

**Comments to the Author**

1. If the authors have adequately addressed your comments raised in a previous round of review and you feel that this manuscript is now acceptable for publication, you may indicate that here to bypass the “Comments to the Author” section, enter your conflict of interest statement in the “Confidential to Editor” section, and submit your "Accept" recommendation.

Reviewer #2: (No Response)

Reviewer #3: (No Response)

2. Is the manuscript technically sound, and do the data support the conclusions?

Reviewer #2: Yes

Reviewer #3: Partly

3. Has the statistical analysis been performed appropriately and rigorously? 

Reviewer #2: Yes

Reviewer #3: No

4. Have the authors made all data underlying the findings in their manuscript fully available?

Reviewer #2: Yes

Reviewer #3: Yes

5. Is the manuscript presented in an intelligible fashion and written in standard English?

Reviewer #2: Yes

Reviewer #3: Yes

6. Review Comments to the Author

Reviewer #2: I was the proverbial reviewer 2 in the previous version of this manuscript. I have read over the new version of the manuscript and reviewed all of the line-by-line responses to my previous comments. As I said previously, this is a great manuscript and well written. The authors have done a great job revising the manuscript. I thank them for their thorough treatment of the reviewer comments. Well done!

Only a couple of minor remaining edits:

85 Change “Due to” to “Because of”. Due to modifies a noun, it is incorrect to use it to modify a verb.

164 The authors might consider using “autumn” instead of “fall”. The latter term is only used in North America.

172 Redundant “diameter at breast height (DBH)” with previous line.

360 Inconsistent missing comma in 5,000

368 Should this be “S3 Appendix” to match the terminology used for S1 and S2, previously? See also line 441.

503 Insert hyphen “field-collected”

506 You need a “respectively” after the statement about large circles are density estimates and error bars are 95% CIs.

Reviewer #3: The authors have a well-written paper about a timely subject. The choice of topics and the broad experimental design seem appropriate to answer their research questions. A red flag I noticed in the methods is that the authors didn’t consider quadratic effects. Considering burn intensity and tree density likely have some optimum values which are neither 0 nor 100%. It seems like there is a strong a priori justification for examining quadratic effects. This choice absolutely needs to be justified. For example white-headed woodpeckers have a negative relationship with tree density, but clearly it cannot have a maximum at 0, as there would be no trees for nesting.

A second red flag is that you argue landscape configuration is likely interacting with forest characteristics to influence results lines 550-587. Why not use fragstats or some other approach to quantifying landscape configuration and incorporate these characteristics among the sites into the modeling approach?

I was thinking about these puzzling methodological choices so I ran supplied example code. The modeling used was more appropriate for an exploratory data analysis, but seems like a pretty flagrant abuse of data dredging best practices. Considering the obvious ignoring of appropriate a priori hypotheses, and instead building gigantic linear combinations of model covariates via a data dredging approach, the authors may benefit from reexamining the methodological tactics employed here.

7. PLOS authors have the option to publish the peer review history of their article (what does this mean?). If published, this will include your full peer review and any attached files.

Reviewer #2: No

Reviewer #3: No

---

## [Author Response · Author response to Decision Letter 1]

23 Apr 2020

Our responses to reviewers are detailed in an attached Word document.

---

## [Editor Report · Decision Letter 2]

28 Apr 2020

Development and evaluation of habitat suitability models for nesting white-headed woodpecker (Dryobates albolarvatus) in burned forest

PONE-D-19-34065R2

Dear Dr. Latif,

We are pleased to inform you that your manuscript has been judged scientifically suitable for publication and will be formally accepted for publication once it complies with all outstanding technical requirements.

With kind regards,

Karen Root, Ph.D.

Academic Editor

PLOS ONE

Additional Editor Comments (optional):

Thank you for the careful attention to all of the suggestions and questions.  The changes you have made enhance the paper providing better balance and clarity, and it is now suitable for publication.  Your paper provides an important addition to our understanding of modeling disturbance effects on vulnerable species.
---

## [Editor Report · Acceptance letter]

1 May 2020

PONE-D-19-34065R2 

Development and evaluation of habitat suitability models for nesting white-headed woodpecker (Dryobates albolarvatus) in burned forest 

Dear Dr. Latif:

I am pleased to inform you that your manuscript has been deemed suitable for publication in PLOS ONE. Congratulations! Your manuscript is now with our production department. 

With kind regards,

on behalf of

Professor Karen Root 

Academic Editor

PLOS ONE